# Computational Prediction of Resistance Induced Alanine-Mutation in ATP Site of Epidermal Growth Factor Receptor

**DOI:** 10.3390/ijms232415828

**Published:** 2022-12-13

**Authors:** Tasia Amelia, Aderian Novito Setiawan, Rahmana Emran Kartasasmita, Tomohiko Ohwada, Daryono Hadi Tjahjono

**Affiliations:** 1School of Pharmacy, Bandung Institute of Technology, Jalan Ganesha 10, Bandung 40132, Indonesia; 2Graduate School of Pharmaceutical Sciences, The University of Tokyo, Tokyo 113-0033, Japan

**Keywords:** EGFR, erlotinib, mutation, prediction, resistance

## Abstract

Epidermal growth factor receptor (EGFR) resistance to tyrosine kinase inhibitors can cause low survival rates in mutation-positive non-small cell lung cancer patients. It is necessary to predict new mutations in the development of more potent EGFR inhibitors since classical and rare mutations observed were known to affect the effectiveness of the therapy. Therefore, this research aimed to perform alanine mutagenesis scanning on ATP binding site residues without COSMIC data, followed by molecular dynamic simulations to determine their molecular interactions with ATP and erlotinib compared to wild-type complexes. Based on the result, eight mutations were found to cause changes in the binding energy of the ATP analogue to become more negative. These included G779A, Q791A, L792A, R841A, N842A, V843A, I853A, and D855A, which were predicted to enhance the affinity of ATP and reduce the binding ability of inhibitors with the same interaction site. Erlotinib showed more positive energy among G779A, Q791A, I853A, and D855A, due to their weaker binding energy than ATP. These four mutations could be anticipated in the development of the next inhibitor to overcome the incidence of resistance in lung cancer patients.

## 1. Introduction

Lung cancer has the largest patient mortality rate in the world and reached 1.796 million cases (18% of total cancer deaths) in 2020 [1,2]. Patients diagnosed with non-small cell lung cancer (NSCLC), which accounts for ~85% of all cases, have a lower 5-year survival rate than the other types [3,4]. Furthermore, the overexpression of the epidermal growth factor receptor (EGFR) is detected in around 60% of NSCLC cases [5]. EGFR is a tyrosine kinase (RTK) receptor in the ErbB family, and the upregulation by the epidermal growth factor (EGF) induces homo and heterodimerization with other members and activates cell signaling through autophosphorylation of the intracellular tyrosine kinase domain [6,7,8].

Several molecules have been developed to target the tyrosine kinase domain of EGFR and have shown favorable outcomes in the early stages of therapy. The tyrosine kinase inhibitors (TKIs) have been approved for clinical use, including first-generation gefitinib [9] and erlotinib [10], second-generation afatinib [11] and dacomitinib [12], as well as irreversible third-generation inhibitor osimertinib [13]. However, drug resistance is developed in most patients after several months due to receptor mutation and limits to the TKI’s effectiveness [5,14,15]. EGFR mutations frequently occur in the tyrosine kinase domain encoded by exon 18–24 (Figure 1). In-frame deletion of exon 19, located between residue 729 and 761, is the most common mutation accounting for 45% of all EGFR in NSCLC cases. The second common mutation is substitution in exon 21, located between residue 824 and 875 within the activation loop, which accounts for 40–45% of all cases. Meanwhile, L858R alteration is the majority case in exon 21 mutation. Substitution in exon 18 and in-frame insertion of exon 20 is less frequent, accounting for 5% of each [5]. The T790M mutation located in exon 20 is detected in 50% of the cases as a second site and has caused resistance in the first-generation TKI therapy [15,16,17].

Patients with rare EGFR mutations have been identified, and clinical research showed a considerable range in TKI’s sensitivity for the mutant EGFR [18]. This research found various residues in the ATP binding site of EGFR with no mutation data in the COSMIC database. The effect of altering those residues was investigated on ATP and TKIs binding in EGFR. The structure of the tyrosine kinase domain of wild-type EGFR in active conformation in complex with erlotinib [19] has facilitated the interaction of the ligands in the mutant EGFR. Therefore, this research aims to acquire molecular insights from the interaction of ligands on EGFR with changes in the conformation and characteristics of the binding site owing to alanine substitution. Significant alterations can serve as an alert for compounds developed as TKIs in the future.

## 2. Results and Discussion

### 2.1. Validation of the Simulation Parameters

The simulation was performed to observe the interaction of the ATP analogue, AMP-PNP (further called ANP), and erlotinib: a first-generation EGFR TKI used clinically to treat lung cancer patients. Three mutated EGFRs were discovered in lung cancer patients with established ATP affinity and were selected as the references to validate the computational method, namely T790M, L858R, and L858R/T790M. ANP interacts with wild-type EGFR by forming hydrogen bonds and several hydrophobic interactions, as shown in Figure 2. The adenosine core can form hydrogen bonds to hinge residue Met793, Cys797, and Gln791, while the phosphate group forms hydrogen bonds to Asp855 and the salt bridge residues Lys745-Glu762. These interactions were also observed in the complex of ANP with T790M and L858R/T790M EGFR but not in the L858R, as seen in Figure 2. The substitution of the hydrophobic side chain of leucine into a charged arginine in the single-mutation L858R causes a change in the binding site, forming an open-formed binding site.

The binding energies were calculated by the MMPBSA method and are shown in Table 1. The ATP analogue was able to bind to the wild-type receptor with an energy of −27.4416 ± 3.3357 kcal/mol (Table 1). This binding energy changed differently when Thr790 and Leu858 were mutated. Single mutation L858R provided the most positive binding energy compared to −15.5423 ± 5.0081 kcal/mol. According to the interaction research, ANP has a decreased interaction with the L858R EGFR, which may explain this result. The more positive the energy, the lower the expected affinity of ATP.

However, Thr790 in a single or secondary mutation of L585R caused only a slight difference with the wild-type complex. Single T790M and double L858R/T790M mutations provide the binding energy of −29.5109 ± 2.9640 and −26.3177 ± 3.8815 kcal/mol, respectively. This finding is consistent with the prior research on the kinetic characterization of the wild-type and mutated EGFR kinases using the Michaelis–Menten constant. Furthermore, the L858R mutant reduces the affinity of ATP and activates EGFR. The T790M mutation restores the ATP affinity to near wild-type levels in both the single and double mutation [20]. Although the EGFR T790M mutant’s kinase activity was comparable to wild-type EGFR, research demonstrated that it might enhance EGFR phosphorylation level over wild-type EGFR [21].

### 2.2. Interaction Study of ATP Analogue in Well-Known EGFR Mutant

The computational study to observe the effect of several mutations of EGFR often discovered in lung cancer patients on ATP binding was conducted. This research used E709A and G719A in exon 18 EGFR while comparing the result to clinical data. Mutation at E709X was reported for 1.5% rare EGFR mutations, while the most common variant was found in E709K and E709A [22]. Although a preclinical study revealed that E709X was significantly less sensitive to first-generation TKIs such as gefitinib and erlotinib [22], the effect on the affinity of ATP to the mutant EGFR binding site remains unknown. By analyzing the interaction during simulation, understanding the mutation effect on ATP binding can be slightly clarified.

The E709A EGFR in complex with ANP and magnesium ion as the cofactor was minimized and prepared, followed by molecular dynamic simulation for 200 ns. The complex was stable and confirmed by the RMSD from around 70 ns until the end of the simulation (Appendix A). The ANP formed hydrogen bonds with Gln791 and Met793, as found in the wild-type complex. Although it lost the interaction with salt bridge residues at the end of the simulation, a new interaction was formed with Ser720, Thr790, and Asp800 through hydrogen binding. This may cause a more negative binding energy than the wild-type, −32.1810 ± 2.6739 kcal/mol (Figure 3, Table 1).

G719X substitution is the most prevalent exon 18 mutations detected in patients, accounting for 1.5–3% of all EGFR mutations in NSCLC patients. G719X is an activating mutation because it is positioned in the P-loop inside the N-lobe and contributes to ATP coordination by arching over the triphosphate moiety and holding the αC-helix in the inactive conformation through hydrophobic contacts. This substitution impairs the hydrophobic interface, resulting in a 10-fold increase in kinase activity relative to the wild-type EGFR [23,24]. The substitution of Gly719 to alanine, aspartate, serine, and cysteine has been reported in NSCLC and G719A, as the frequent mutation was chosen to be computationally studied [25]. Kobayashi et al. found that G719A demonstrated a poor response to first-generation TKIs but was sensitive to the second-generation TKIs afatinib and neratinib [22].

This research observed the interaction of ANP and G719A to clarify the cause of TKIs resistance. ANP occupied the inner binding pocket known as the allosteric pocket due to the hindrance by the side chain elongation of residue 719 in the P-loop. At the end of the simulation, the nitrogen atoms of the adenine core formed hydrogen bonds to Thr854 and Asp855 with the hydrophobic interaction of Cys755 and Leu844. The phosphate group keeps ANP binding in the pocket by forming a hydrogen bond with Leu718, Asp800, and Glu804. Unlike the other EGFR mutants, the ribose group of ANP formed a hydrogen bond with the hinge residue Met793 instead of the adenine. This interaction causes ANP to bind better to its binding site with more negative MMPBSA energy than wild-type EGFR, −32.9977 ± 3.4104 kcal/mol.

### 2.3. Binding Study of ATP Analogue in Novel EGFR Mutations

Alanine scanning was performed on the wild-type structure to identify the role of the specific residue in the stability of the protein and its interaction with the ligand [26]. Residues in the ATP binding site with no single mutation or resistance data in the COSMIC database were selected to be substituted with alanine. Furthermore, 25 residues in exons 18–21 (Figure 4) were used and provided 25 single-mutated EGFR. The non-hydrolysable analogue of ATP, adenylyl-imidodiphosphate (ANP), and the first-generation TKIs erlotinib were used as the ligands. Mutated EGFR was minimized to determine the stable conformation of the protein, and the ligands were docked to the ATP-binding site. The conformation with the best docking score was used as the initial pose to be further studied using molecular dynamics for 200 ns. The binding energy and interaction of the ligands against the EGFR were monitored as the research parameter. The stability of the protein backbone was monitored during simulation using the RMSD graph (Appendix A).

The position of Mg was retained from the ANP phosphate groups in the optimum position based on the previous research [27], and an optimized structure was used for the interaction study. Based on the result of the molecular dynamic simulation, all complexes of the mutated EGFR with ANP showed stable conformation during the last 50 ns simulation. The binding energies of the ligand against 25 mutated EGFR were compared to the wild-type complex, as shown in Table 2. The positive binding energy difference (ΔΔG) means that the ANP affinity is predicted to be lower, while negative ΔΔG predicts a higher affinity to the mutated receptor.

The mutation of Glu762 to alanine showed the highest positive ΔΔG compared to the others. This residue is one of the key amino acids that form a salt bridge with positively charged Lys745 by forming an ionic interaction [28]. The negatively charged side chain in alanine caused a conformation change in the ATP binding site and weakened the ANP, as shown in Figure 5a. The phosphate group that was supposed to interact with Glu762 in the wild-type EGFR could not form the hydrogen bond to the αC-helix and was stepped out from the binding site at around 140 ns of the simulation. However, it remained in the outer part of the binding site until the end of the simulation. Most key interactions were not found in the last 50 ns simulation of the ANP–E762A complex (Figure 5b). The K745A mutation showed only a slight positive difference in the binding energy and preserved the interaction with the key amino acids Met793 and Cys797. This mutation resulted in unstable receptor conformation even when the simulation was prolonged to 300 ns.

The mutation of Ile789 to alanine was also analyzed to strengthen the argument from the positive difference obtained. Furthermore, the mutation of I789A reduces the size of the binding site, indicating the ANP tends to leave and bind on the outside, as in E762A. Most of the ANP interactions with the wild-type EGFR could not be found in the I789A–ANP complex. These two positive results indicate that some mutations are predicted to cause ATP to lose affinity for EGFR due to changes in the shape of the binding site.

On the other hand, eight single-mutations showed ΔΔG to be more negative than ANP against wild-type EGFR, including G779A, Q791A, L792A, R841A, N842A, V843A, I853A, and D855A. The I853A complex with ANP showed the most negative binding energy with ΔΔG of −9.4553 kcal/mol. During the mutation, ANP could bind stably and form hydrogen bonds to Ser720, Thr790, Gln791, Met793, Cys797, and Asp800 (Figure 6). Moreover, G779A, the second most negative difference, showed similar results. ANP could occupy the G779A binding pocket stably during simulation while maintaining the interaction between its adenine core and several pocket residues such as Thr790, Gln791, and Met793. The ribose group formed a hydrogen bond with Arg841 rather than interacting with Cys797 as in the ANP–I853A complex. The phosphate groups lost the hydrogen bond with Ser720 but were replaced by two new hydrogen bonds with Leu718 and Cys797 (Figure 6c).

Negative ΔΔG was predicted to cause an increase in ATP affinity, decreasing the inhibitor due to the binding competition or resistance. This result has to be further confirmed by the in vitro assay and may become a concern to developing TKIs.

### 2.4. Binding Study of Erlotinib in Mutated EGFR

Activating mutations in the EGFR kinase domain have been found in between 10% and 60% of NSCLC patients, where exon 19 deletion and the point mutation L858R accounted for roughly 85% of the mutant NSCLC [29]. These classical activating mutations were more sensitive to EGFR inhibitor therapy than the wild-type kinase [30,31,32,33]. Erlotinib is the first-generation TKI to be approved to treat patients with advanced or metastatic NSCLC with EGFR-activating mutations [34]. Compared to conventional chemotherapy, it delays symptom progression, improves the quality of life, and prolongs survival with EGFR exon 19 deletions or L858R substitution mutations [35,36]. However, most patients showed reoccurring cancer during kinase inhibitor treatment, and approximately half of them had a drug-resistant EGFR T790M mutation, particularly with first-generation TKIs [15,33,34,35]. Less common EGFR kinase domain mutations have also been increasingly found in NSCLC patients [37]. Erlotinib is ineffective in most cases due to inhibitor resistance, resulting in a relapse in lung cancer patients [38,39].

Several novel mutations without COSMIC data were designed, and the binding energy and interaction of erlotinib to the mutated EGFR compared to the wild-type EGFR was analyzed. Erlotinib could bind stably to the ATP binding site in wild-type EGFR with ΔG of −33.5815 ± 4.4701 kcal/mol (Table 3). This forms a hydrogen bond to Met793, the key hinge region residue in TKIs interaction [40]. Moreover, it also retains several hydrophobic interactions with Ala743, Cys797, and Leu844 during simulation (Figure 7). The binding energy is more negative than the ANP in wild-type EGFR, suggesting that erlotinib can compete well with ANP. This is consistent with prior findings, which found that it had good effectiveness in the EGFR mutation-negative case while providing superior outcomes in the EGFR mutation-positive case [41].

All 25 single-mutated EGFRs were combined with erlotinib, and the molecular dynamics simulations were performed for 200 ns of the complex. This showed stable conformation during the last 50 ns simulation (Appendix A), and their binding energies were calculated using the MMPBSA for the last 50 ns. In general, mutant complexes showed more positive binding free energies than wild-type EGFR, with the most positive values belonging to the V742A–erlotinib complex (Table 3).

A potent TKI is expected to bind competitively with ATP in the binding region of the receptor even though there is an increase in the ATP affinity for the mutant receptor. According to prior research (Section 2.3), eight mutations are anticipated to improve the ATP affinity, namely G779A, Q791A, L792A, R841A, N842A, V843A, I853A, and D855A, with ANP binding free energies of −35.6042 ± 3.2002, −34.8260 ± 2.9055, −30.3337 ± 3.0148, −29.2160 ± 3.3257, −29.4135 ± 2.2418, −29.0340 ± 2.3082, −36.8969 ± 3.2786, and −28.6598 ± 2.8774 kcal/mol, respectively (Table 2). These energies were compared with the binding energy of erlotinib to the receptor. The binding energies of erlotinib were found to be more positive to four mutated EGFRs, namely G779A, Q791A, I853A, and D855A, compared to the ANP binding energies of those mutated residues.

Even though erlotinib could retain the interaction with the key residue Met793 in all four mutations, several hydrophobic interactions were lost with the binding pocket residues. The G779A–erlotinib complex with more positive energy indicates a loss of interaction with Val726, Met766, Leu777, and Leu788, which was found in the wild-type complex interaction (Figure 7). The Q791A, I853A, and D855A mutants also exhibited a similar occurrence in which their hydrophobic interactions were lost with Met766-Leu788, Val726-Leu777, and Leu777-Leu788, respectively (Figure 8).

The interaction with the residues was shown in several crystal structures of EGFR with the first- and fourth-generation TKIs, which are allosteric inhibitors [42]. Although no research focused on the role of hydrophobic interactions with Val726, Met766, Leu777, and Leu788, the binding energies that were obtained indicated the loss effect of these interactions on ligand binding to receptors.

From these results, it was predicted that erlotinib could have resistance problems when the patient overexpression of G779A, Q791A, I853A, and D855A EGFR. Therefore, the development of new inhibitors for EGFR is expected to anticipate resistance events by maintaining the affinity for these four mutated EGFRs. However, the mutation of the residues only to alanine with methyl sidechains is believed to be the limitation of this study. Further analysis using the other 18 possible residues has to be further performed to determine the effect of residue properties on the binding ability of ATP and TKI in the receptor.

## 3. Materials and Methods

### 3.1. Macromolecule Preparation

The crystal structure of wild-type EGFR was retrieved from the RCSB Protein Data Bank with PDB ID of 1M17 (EGFR tyrosine kinase domain with 4-anilinoquinazoline inhibitor erlotinib) [19]. The protein was prepared by removing water molecules and crystallized ligands, adding polar hydrogen atoms, and directly calculating the Kollman charges for the wild-type EGFR simulation.

Meanwhile, 25 single-substituted EGFR was prepared by converting the residues to alanine, namely Val726, Tyr727, Pro741, Val742, Lys745, Arg748, Glu762, Cys775, Leu777, Leu778, Gly779, Leu788, Ile789, Gln791, Leu792, Met793, Phe795, Arg841, Asn842, Val843, Leu844, Val845, Lys852, Ile853, and Asp855. As the reference, the T790M, L858R, L858R/T790M, E709A, and G719A mutation, known to cause resistance to TKIs, were also used to validate the in silico method performed. Alanine substitution was conducted using BIOVIA Discovery Studio [43], and the mutated EGFRs were minimized using Amber16 to obtain a stable conformation of each macromolecule.

### 3.2. Ligand Preparation

The ATP analogue, ANP, and the first-generation TKI erlotinib were used as the ligand for EGFR. Furthermore, the ANP crystal and the magnesium ion were retrieved from the 5D41 crystal structure [44]. The three-dimensional structure of erlotinib was designed and optimized using GaussView 5.0.8 and Gaussian09 [45] with the density functional theory (DFT) method and 6–31G basis set. For validating the docking simulation parameters, crystallized ligand erlotinib was separated from the complex using BIOVIA Discovery Studio in the re-docking process.

### 3.3. Docking Simulation

The docking simulation was carried out using AutoDockTools 4.2 and MGLTools 1.5.6 [46]. Before the simulation, the ligand preparation was performed by setting the rotatable bonds and adding Gasteiger charges. EGFR kinase as the macromolecule was prepared by adding polar hydrogen to the protein. The site-specific docking towards known residues of the EGFR binding site was performed using erlotinib as a ligand to obtain the pocket. This formed a grid size of 40 × 50 × 40 with a spacing of 0.375 Å, covering the entire ATP-binding site. EGFR kinase and ligands were rewritten in the PDBQT format, and the docking procedure was fixed for the protein and was flexible for the ligands. The electrostatic and affinity maps for all the atom types present were calculated. Meanwhile, the docking was performed using the Lamarckian genetic algorithm (GA 4.2) with 100 randomized conformational searches and a maximum evaluation number of 2,500,000. Crystallized erlotinib (AQ4) in the 1M17 structure was redocked to the binding site to validate the accuracy of the protocol, and the RMSD value was calculated. The conformation of the minimum energy was chosen as the input for molecular dynamic simulation.

### 3.4. Molecular Dynamic (MD) Simulation

The simulation was performed using the Amber 16 package [47], and the molecules were generated as preparation files with antechamber and parmchk modules [48,49]. A molecular dynamic (MD) simulation was performed at 200 ns for wild-type and mutant EGFR with ANP or erlotinib using the general amber force field (GAFF) and FF14SB [49,50]. Furthermore, the magnesium ion (Mg^2+^) force field parameter was applied from the TIP3P water model [51]. The complexes were then protonated, and the overall charge was neutralized by adding a suitable counter ion (Na^+^). The TIP3PBOX water model was utilized to solvate the molecules, with the edge of the box at 10 Å from the solute, and the simulation was run with each complex system maintained at 310 K under constant pressure.

Energy minimization was conducted for 5000 steps using the particle mesh Ewald molecular dynamics (PMEMD) of Amber16. The minimized systems were heated to 310 K, and all restraints applied were released until equilibrium. The simulations were run for 200 ns, and the trajectories were collected every 10 ps to obtain insights into the interactions at the atomistic level. Meanwhile, the root mean square deviation (RMSD) and hydrogen bond average was calculated by the CPPTRAJ module of the Amber tool. Finally, the mmpbsa.py module was used to determine the binding energy of each complex using the MD trajectories [52]. Binding energies were obtained using 500 extracted snapshots from the final 10 to 50 ns trajectory of 200 ns simulation data. The internal dielectric constant of 4.0 was incorporated into the input bond energy calculation using MMPBSA for complexes with ANP-Mg [53].

## 4. Conclusions

The alanine mutation scanning followed by the molecular dynamic simulation was performed to predict the effect of residue mutation on the binding affinity of ligands against EGFR, which led to drug resistance. The substitution of the residue to alanine was performed and provided additional protein stability and erlotinib interaction in the binding site, diminishing or enhancing the inhibitory activity. Based on the binding energy calculated, several mutations of EGFR in complex with the ATP analogue were predicted to enhance the ATP affinity and alter the effectivity of TKIs, namely G779A, Q791A, L792A, R841A, N842A, V843A, I853A, and D855A. Due to the loss of hydrophobic interactions, erlotinib showed a greater positive binding energy to G779A, Q791A, I853A, and D855A. This in silico research predicted that the four mutations had the potential to cause resistance to the TKIs and can be anticipated in developing new inhibitors in the future. The research can be broadened by investigating four other potential mutations in the four residues and conducting in vitro studies to verify the role of those mutations on drug resistance.

## Figures and Tables

**Figure 1 ijms-23-15828-f001:**
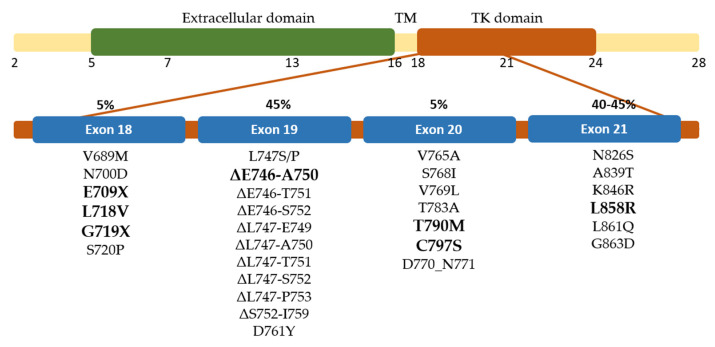
Distribution of exons encoded EGFR and the mutation site of EGFR in NSCLC patients. The most common mutations in each exon are shown in bold.

**Figure 2 ijms-23-15828-f002:**
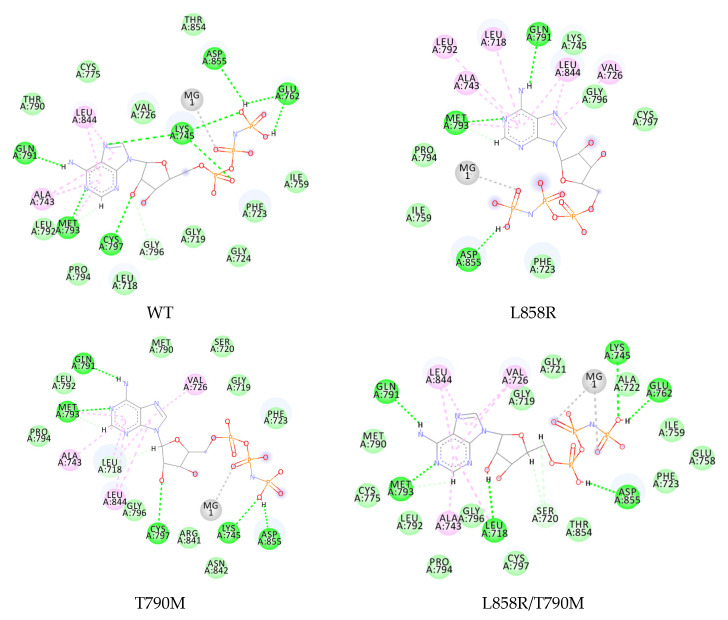
Two-dimensional interaction of ANP in the wild-type and mutated EGFR. Interactions are colored based on the type: 
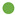
, hydrogen bond; 
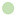
, van der Waals; 
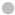
, metal acceptor; 
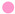
, pi-alkyl.

**Figure 3 ijms-23-15828-f003:**
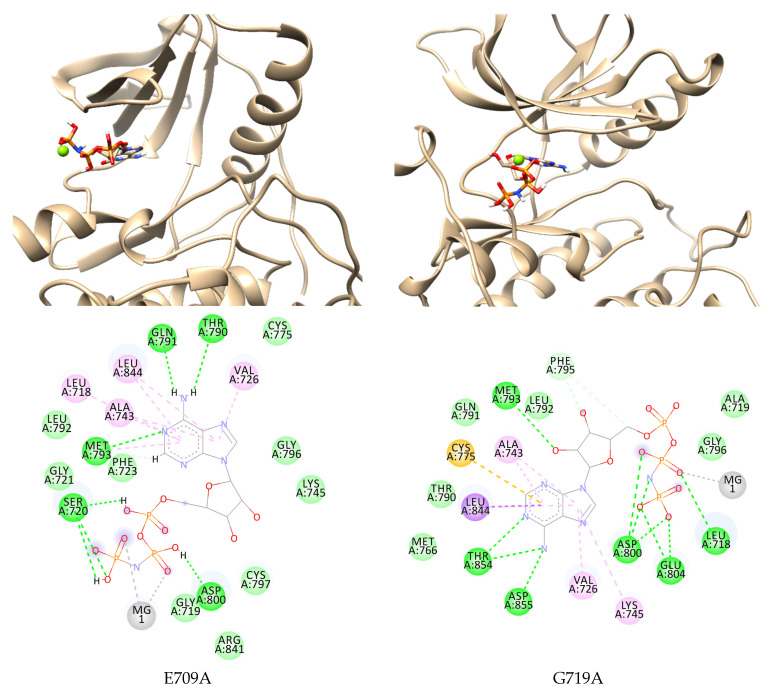
Two- and three-dimensional interaction of ANP in E709A and G719A EGFR. Interactions are colored based on the type: 
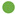
, hydrogen bond; 
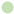
, van der Waals; 
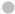
, metal acceptor; 
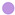
, pi-sigma; 
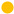
, pi-sulfur; 
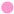
, pi-alkyl.

**Figure 4 ijms-23-15828-f004:**
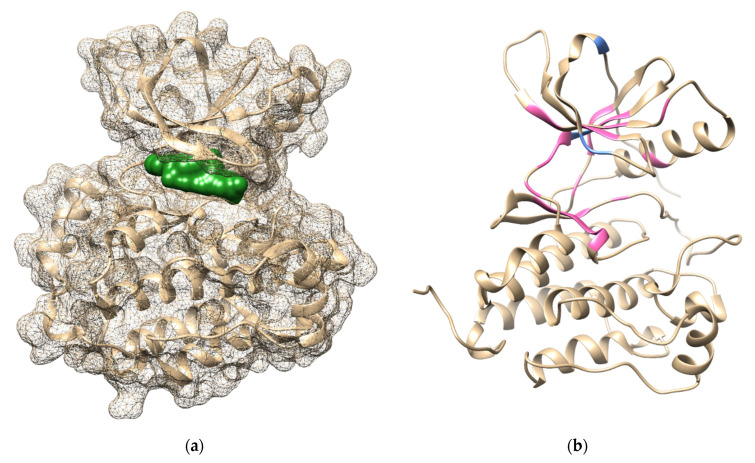
(**a**) Binding pocket of ANP-Mg in EGFR based on 5D41 crystal structure; (**b**) Mutation position of the reference mutation (blue) and test mutation (pink) on 1M17 crystal structure.

**Figure 5 ijms-23-15828-f005:**
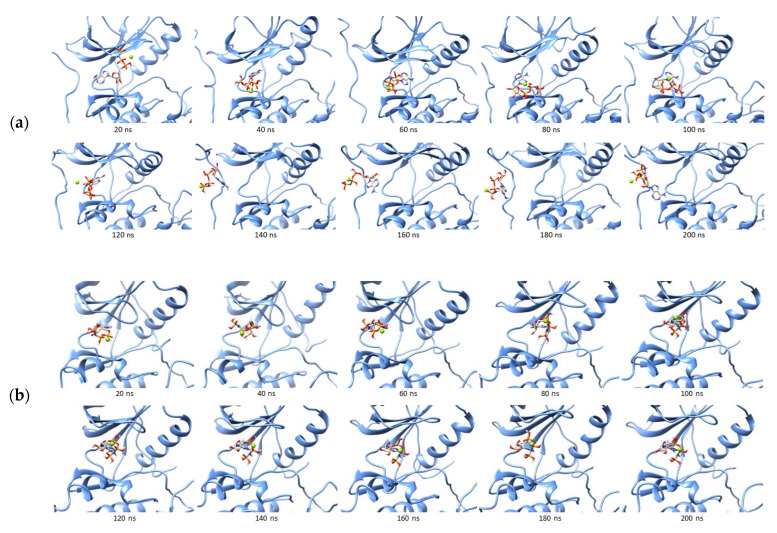
Trajectories of ANP during 200 ns simulation in the binding site of E762A (**a**), K745A (**b**), I789A (**c**), and (**d**) Two-dimensional interaction of ANP to the mutated receptor: 
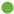
, hydrogen bond; 
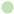
, van der Waals; 
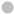
, metal acceptor; 
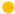
, pi-sulfur; 
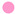
, pi-alkyl.

**Figure 6 ijms-23-15828-f006:**
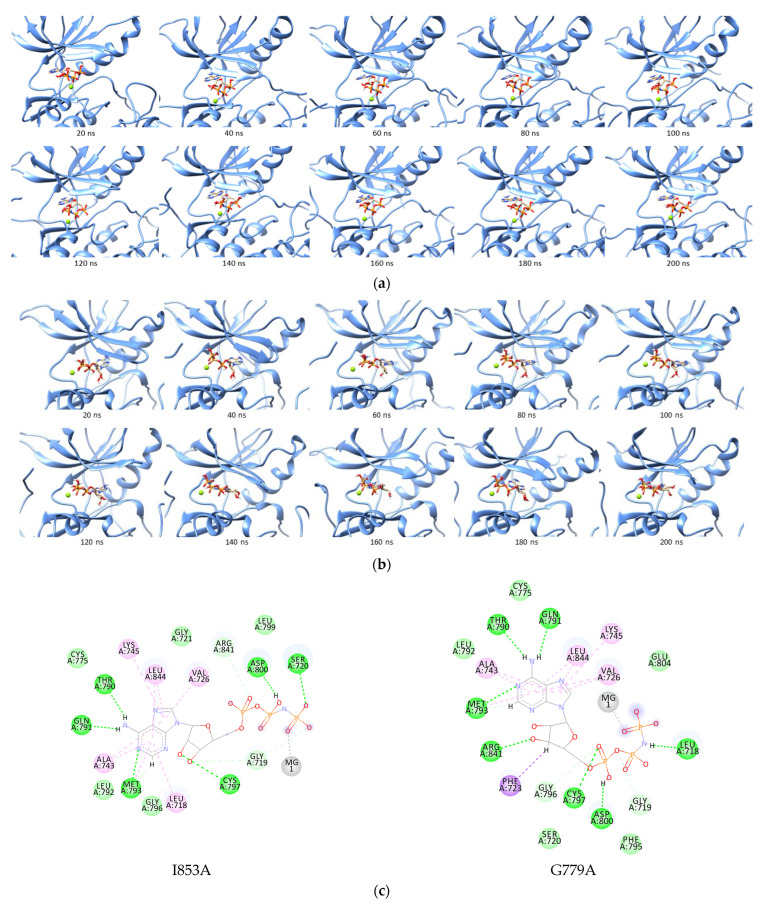
Trajectories of ANP in the I853A (**a**) and G779A (**b**) binding site during simulation, and (**c**) 2D interaction of ANP to the I853A and G779A receptor: 
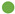
, hydrogen bond; 
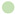
, van der Waals; 
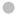
, metal acceptor; 
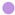
, pi-sigma; 
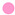
, pi-alkyl.

**Figure 7 ijms-23-15828-f007:**
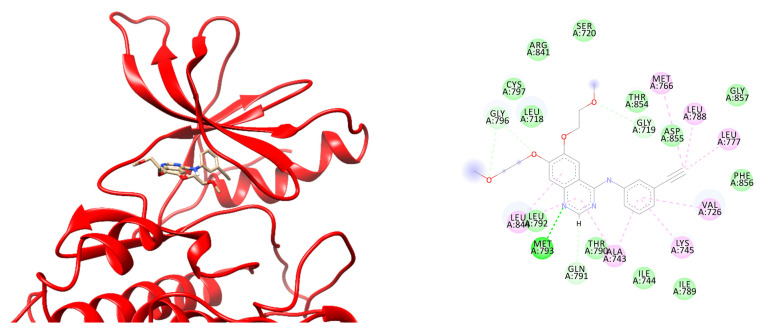
The position of erlotinib in ATP binding site of EGFR (**left**) and two-dimensional interaction of erlotinib to wildtype EGFR (**right**): 
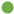
, hydrogen bond; 
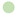
, van der Waals; 
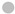
, metal acceptor; 
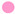
, pi-alkyl.

**Figure 8 ijms-23-15828-f008:**
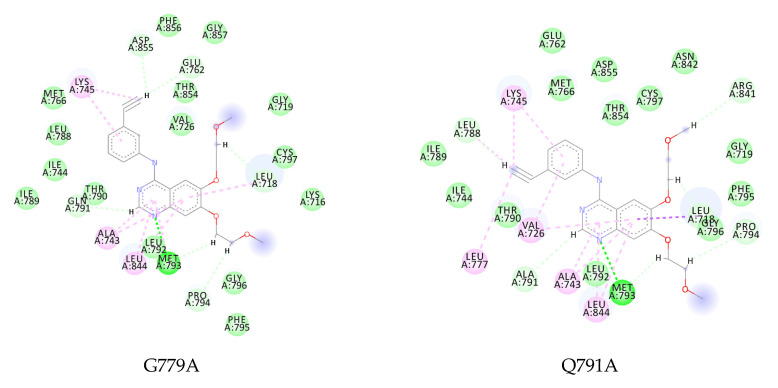
Two-dimensional interaction of erlotinib to G779A, Q791A, I853A, and D855A (
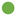
, hydrogen bond; 
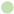
, van der Waals; 
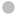
, metal acceptor; 
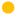
, pi-sulfur; 
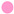
, pi-alkyl).

**Table 1 ijms-23-15828-t001:** Binding energy of ATP analog to the wild-type and known mutated EGFR.

Ligand	EGFR	ΔG (kcal/mol)	SD	ΔΔG to Wildtype
ATP	Wild-type	−27.4416	3.3357	0
	T790M	−29.5109	2.9640	−2.0693
	L858R	−15.5423	5.0081	11.8993
	L858R-T790M	−26.3177	3.8815	1.1239
	E709A	−32.1810	2.6739	−4.7394
	G719A	−32.9977	3.4104	−5.5561

**Table 2 ijms-23-15828-t002:** Free binding energy of ATP to mutated EGFR.

EGFR	ΔG *		SD	ΔΔG **	EGFR	ΔG *		SD	ΔΔG **
V726A	−24.5951	±	6.8073	2.8465	Q791A	−34.8260	±	2.9055	−7.3844
Y727A	−26.5902	±	2.9467	0.8514	L792A	−30.3337	±	3.0148	−2.8321
P741A	−23.3618	±	4.0048	4.0798	M793A	−26.1200	±	3.1922	1.3216
V742A	−24.4672	±	4.1159	2.9744	F795A	−20.9834	±	3.6914	6.4582
K745A	−27.2926	±	2.9010	0.1490	R841A	−29.2160	±	3.3257	−1.7744
R748A	−25.5706	±	5.4211	1.8710	N842A	−29.4135	±	2.2418	−1.9719
E762A	−4.2624	±	4.7396	23.1792	V843A	−29.0340	±	2.3082	−1.5924
C775A	−15.0233	±	4.9797	12.4183	L844A	−26.3112	±	3.7357	1.1304
L777A	−8.3795	±	4.8769	19.0621	V845A	−18.5851	±	4.6366	8.8565
L778A	−24.5564	±	3.7146	2.8852	K852A	−23.0544	±	4.1278	4.3872
G779A	−35.6042	±	3.2002	−8.1626	I853A	−36.8969	±	3.2789	−9.4553
L788A	−11.6746	±	3.5942	15.7670	D855A	−28.6598	±	2.8774	−1.2182
I789A	−9.6311	±	6.2091	17.8105					

* in kcal/mol, ** compared to the binding free energy in wild-type EGFR.

**Table 3 ijms-23-15828-t003:** Binding free energy (kcal/mol) of erlotinib to the wild-type and mutated EGFR TK.

EGFR	ΔG *		SD	ΔΔG **	EGFR	ΔG *		SD	ΔΔG **
**Wild-type**	**−33.5815**	**±**	**4.4701**	**0**	I789A	−30.2833	±	4.1253	3.2982
V726A	−25.3940	±	4.2140	8.1875	Q791A	−29.8615	±	4.3156	3.7200
Y727A	−31.0579	±	4.7024	2.5236	L792A	−31.7232	±	4.1862	1.8583
P741A	−31.2215	±	4.1350	2.3600	M793A	−25.6215	±	4.1862	7.9600
V742A	−19.3823	±	6.4425	14.1992	F795A	−31.0544	±	4.1622	2.5271
K745A	−27.3411	±	5.1047	6.2404	R841A	−33.6124	±	4.5257	−0.0309
R748A	−26.3685	±	3.9220	7.2130	N842A	−29.6648	±	4.0082	3.9167
E762A	−29.8702	±	3.9274	3.7113	V843A	−32.1735	±	4.7292	1.4080
C775A	−29.7660	±	4.0230	3.8155	L844A	−30.1431	±	3.4659	3.4384
L777A	−28.8312	±	3.7831	4.7503	V845A	−32.0817	±	4.6266	1.4998
L778A	−29.5240	±	4.2194	4.0575	K852A	−29.8120	±	4.2821	3.7695
G779A	−27.8010	±	4.3619	5.7805	I853A	−32.5858	±	4.1526	0.9957
L788A	−28.4853	±	5.4903	5.0962	D855A	−27.9652	±	3.7118	5.6163

* in kcal/mol, ** compared to the binding free energy in wild-type EGFR.

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
