# Peer review of "Computational Prediction of Resistance Induced Alanine-Mutation in ATP Site of Epidermal Growth Factor Receptor"

_ijms, 2022, doi:10.3390/ijms232415828_

Round 1

Reviewer 1 Report

This MS studies the molecular interactions of ligands on EGFR with AA changes via in silico mutagenesis. I have just two comments:

Why the authors used ANP instead of ATP?

Check: P4-L120:  Thr7790

Reviewer 2 Report

The following comments should be taken into account before accepting the manuscript: 

• Avoid using acronym in the abstract, which may confuse readers.  • There is a lack of RMSD figures, which must be added in the manuscript.  • Line 207: “G779A, […] , showed similar results”. The author must explain why G779A showed “similar results” to I853A since the docking score, interaction mode of two mutations are not similar.  • Line 257 – 266: Why did the study said 4 mutations G779A, Q791A, I853A, D855A had more positive values than ANP. The ΔG of erlotinib - mutant receptor > ΔG of ANP - mutant receptor • Line 267 – 279: It is unappropriate to discuss the results throughout Fig 8 and Fig 7, since the figure 7 describes interaction between erlotinimib and the WT ptotein. Moreover, the information given from line 267 to 269 is not appropriate with Fig 8.  • In conclusion, the authors should discuss and comment on continuing studies.  • There are some contents in “Results and Discussion” related to theory or manipulation should be replaced in the “Materials and Methods”.  • The authors should orgainize the figures and tables in correct format. Figure 6 should be added caption in picture (b).   • There are some grammar and vocab errors. This manuscript should be carefully revised by a native speaker.

Reviewer 3 Report

Authors have predicted drug resistance mutations in ATP binding sites of EGFR using in silico approach. Authors have done extensive study of docking, MD simulations and binding free energy calculations with alanine scanning. Although authors have presented data accurately and concisely, there are very important points that need to be addressed.   1) Usually alanine scanning is used to identify binding sites or to identify residues which are important for the binding based on binding free energy change value. Authors have identified drug resistance mutations where mutations are only to alanine. There are other 18 possible mutations for each residue that needs to be explored. Without that it is hard to say this work is predicting drug resistance mutations. Authors should have a look at these kind of paper https://doi.org/10.3390/molecules27175413 2) Authors have simulated complexes for 200ns but for MMPBSA calculations, they have used last 10 ns trajectory data which is very small. Authors have also looked at the RMSD and stability of complexes, where some were stable after 70ns. so  authors should take after 70 ns trajectory data to calculate MMPBSA so that way all different conformations will be studied . In the last 10ns complex might be trapped in one local energy minima but there other local and global energy minima need to be studied. Lat 10 ns conformations may not be the bioactive conformations.

Round 2

Reviewer 2 Report

The revised version of the manuscript is improved. I recommend the publication of the manuscript in this current form.

Author Response

Dear Reviewer,

Thank you very much for accepting our revised manuscript.

Sincerely yours,

dh tjahjono

Reviewer 3 Report

Find the attached response.

Round 3

Reviewer 3 Report

Authors have addressed all the relevant concerns. I do not have any other comments.